# Implantable Immunosuppressant Delivery to Prevent Rejection in Transplantation

**DOI:** 10.3390/ijms23031592

**Published:** 2022-01-29

**Authors:** Madonna Rica Anggelia, Ren-Wen Huang, Hui-Yun Cheng, Chih-Hung Lin, Cheng-Hung Lin

**Affiliations:** Center for Vascularized Composite Allotransplantation, Department of Plastic and Reconstructive Surgery, Chang Gung Memorial Hospital, Chang Gung Medical College and Chang Gung University, Taoyuan 333, Taiwan; mranggelia@yahoo.com (M.R.A.); aaronhuang0327@gmail.com (R.-W.H.); hycheng21@gmail.com (H.-Y.C.)

**Keywords:** immunosuppressant, rejection, transplantation, implantable drug delivery

## Abstract

An innovative immunosuppressant with a minimally invasive delivery system has emerged in the biomedical field. The application of biodegradable and biocompatible polymer forms, such as hydrogels, scaffolds, microspheres, and nanoparticles, in transplant recipients to control the release of immunosuppressants can minimize the risk of developing unfavorable conditions. In this review, we summarized several studies that have used implantable immunosuppressant delivery to release therapeutic agents to prolong allograft survival. We also compared their applications, efficacy, efficiency, and safety/side effects with conventional therapeutic-agent administration. Finally, challenges and the future prospective were discussed. Collectively, this review will help relevant readers understand the different approaches to prevent transplant rejection in a new era of therapeutic agent delivery.

## 1. Introduction

Innovations in immunosuppressant delivery to prevent allotransplant rejection have grown rapidly in recent decades. The immunosuppressant carriers should ideally meet two prerequisites. First, they should deliver the immunosuppressant in a controlled dose directed to the organ target during the treatment period. Second, they should facilitate the active entity of the immunosuppressant to the site of rejection. Conventional systemic drug administrations, such as oral intake, intravenous, intraperitoneal, intramuscular, and subcutaneous injections, are unable to meet any of these prerequisites. To circumvent this, biodegradable polymers (such as hydrogels, micro/nanoparticles, and microspheres) for biomaterial formulation have been developed to provide advantages for immunosuppressant delivery, including the ability to support the local delivery of the drugs at a constant rate, to minimize the dosage requirement and then further enhance drug efficacy, and prevent adverse effects. Moreover, these polymers have good stability and high biocompatibility. Many reports have applied the biodegradable polymers as implantable immunosuppressant delivery carriers and show potential in suppressing transplantation rejection. Herein, we will review the current application of implantable immunosuppressant delivery that employ biodegradable polymers to prevent rejection in allotransplantation.

## 2. Immunosuppressants to Prevent Allograft Rejection in Transplantation

Transplantation is used to transfer an organ, a tissue, or a group of cells from the donor to the recipient (or host) or to different sites in the same individual [1], with the purpose of saving lives or improving the quality of life through the replacement of damaged organs or tissue with functional ones [2,3]. However, rejection caused by the direct or indirect immune mechanism of the recipient is one of the main obstacles that occur after transplantation [4].

In a transplantation setting, both innate [5] and adaptive immune cells [6,7], such as macrophages, natural killer (NK) cells, helper T (Th) cells (CD4^+^ T cells), cytotoxic T cells (CD8^+^ T cells), B cells, as well as cytokines and antibodies are involved in the direct or indirect immune mechanism of allograft rejection [8]. To inhibit effector cells activation/proliferation or enhance regulatory cells activation/proliferation [9], the common approaches to immunosuppression include costimulation blockades to block dendritic cells (DCs)–T cells interactions [10], anti-inflammatory recombinant proteins [11,12], immunosuppressive drugs [13], and anti-inflammatory cytokines derived from immunomodulatory stem cells, such as mesenchymal stem cells (MSCs) [14,15].

The early development of immunosuppressants, such as cyclosporine (CsA), tacrolimus (TAC), and rapamycin (RAPA), targeted the T cells pathways [16]. This is because recipient T cells play a significant role in acute/chronic rejection [17], with direct recognition of foreign antigens that are presented by the receptors of donor and recipient dendritic cells [18,19]. Once T cells are activated, they proliferate and secrete cytokines that regulate other immune responses, including B cells [20] that can produce an alloantibody that is detrimental to long-term graft survival [21]. Mycophenolate mofetil (MMF) has been applied to inhibit B cell proliferation [22]. Later on, more immunosuppressant types were developed. For instance, tofacitinib that inhibits the immune response through the janus kinase pathway, monoclonal antibodies (clazakizumab, tocilizumab) to inhibit inflammatory cytokines (such as interleukin (IL)-6), and multimeric Fas ligands (Adipogen) that can improve the survival of MSCs survival and support MSCs to induce T cells apoptosis (Figure 1) [23,24,25,26,27].

Broadly speaking, the transplant recipients may receive pharmacological treatments during the induction, maintenance, or rejection phases [28,29] based on the post-transplant time and the specific infection/rejection profile status [13,30]. Treatment during the induction phase aims to immediately eradicate the immune response, while treatment during the maintenance phase seeks to maintain the tolerant condition in which the immune system is sufficiently suppressed to minimize allograft rejection but is still able to respond to infections [31]. Multiple or single immunosuppressive drug administration is commonly applied in the induction or maintenance phase. Although not always successful, combination therapy that is intended to deplete B cells is used in the rejection phase. Costimulation blockades, T cells or lymphocytes depletion, and immunomodulatory cells are also used during the induction phase because immunosuppressive drug administration alone is not enough [32].

## 3. Types of Implantable Immunosuppressant Delivery System in Transplantation

Implantable immunosuppressant delivery systems have been devised to localize drug delivery and minimize the side effects of post-transplantation therapies. Implantable immunosuppressant delivery can be made through various systems, such as hydrogels, nano/microparticles [33], micelles [34], microspheres [33], and liposomes [35] (Figure 2). Several implantable immunosuppressant deliveries applied in transplantation are summarized below.

### 3.1. Hydrogel

A hydrogel has hydrophilic properties that are constructed from a 3D network polymer, is able to swell under water, and retains a large amount of water. It is feasible for immunosuppressant delivery because their unique physical characteristic to transform from liquid into a gel due to environmental changes, such as physical stimuli (temperature, pressure, and light) and chemical stimuli (pH, ion, and solvent composition) [36]. Hydrogels for immunosuppressant delivery can be made from polymers, such as poly-d,l-lactic-co-glycolic (PLGA), polyethylene glycol (PEG), and triglycerol monostearate (TGMS) [37,38]. They could be homopolymers, copolymers, or multipolymers [37,38,39]. The incorporation of peptides or nanoparticles may enhance their function [40,41].

### 3.2. Microsphere

Double-walled polymer microspheres, with a 1–1000 μm diameter size, could be synthesized by emulsification [42]. Scientific experts commonly employ biodegradable polymers, poly(lactic-co-glycolic acid), and poly(l-lactic acid) to make microspheres to control drug release [43,44,45]. Their low toxicity and degradation are suitable for use as a drug delivery system without side effects. This system has been utilized to treat various diseases successfully when inconvenience and limitations related to chronic oral treatments cannot be avoided [46]. Another application is the encapsulation of tacrolimus (TAC), which is effective in controlling rejection in the porcine small bowel transplantation and improving the allograft survival time in a heart transplant rat model while avoiding side effects [47,48].

### 3.3. Nanoparticles

Nanoparticles have been extensively studied as delivery vehicles for therapeutics. They are called nano since they have a 1–100 nm diameter size [49]. Because of their excellent biocompatibility and controllable biodegradability, PLGA nanoparticles prevent macromolecules from instant degradation in vivo. Hence, drug encapsulation by PLGA nanoparticles improves stability. Additionally, due to their size, these particles can target and access many parts of the body and can also penetrate specific tissues through the fenestrations in the endothelium of inflamed tissues or receptors for overexpressed targeted cells [50]. Better efficacy than soluble drugs results in the minimum required drug dosage that should be implemented [51]. The application of PLGA nanoparticles to prolong the allograft survival has been reported [52,53].

### 3.4. Micelles and Liposomes

Micelles and liposomes are vesicular structures composed of lipids that are formed in aqueous solutions. Unlike liposomes composed of lipid bilayers, polymer-based micelles comprise only closed lipid monolayers that are hydrophilic on the polar side and have fatty acids on the core side (Figure 2). Previously, both have also been used in various applications, such as drug delivery, hormone and chemotherapy, and vaccine delivery. In a transplantation study, Kuppan et al. optimized the dose of an immunosuppressant that attenuated naïve T cell proliferation and differentiation through CD28 costimulation [54] (dexamethason (Dex)) released with the micelles delivery system in the islet allograft. Nanosized micelles have also been reported to be applied in cornea transplantation, with good cytoprotective capacity and safety profile [55,56,57].

### 3.5. Scaffold

The scaffold delivery system can be implantable or injectable [58]. Scaffolds can be synthesized from polymerized alginate, were usually employed to delivery cells, and improved survival, such as MSCs in wound healing [59]. Scaffolds can retain the nanoparticle mobility. The quick release of nanoparticles from scaffolds directly reduces the effects of the immunosuppressive drug depot. Li et al. synthesized a scaffold delivery system to maintain the delivery of growth factors and immunosuppressant-loaded nanoparticles to enhance the survival rate of transplanted stem cells. They used a self-assembled peptide (RADA16) as a basic unit of hydrogels that could anchor and immobilize the nanoparticles; therefore, it can localize and sustain drug release for 28 days and ultimately enhance the survival of transplanted cells [41]. Mayorga et al. designed a 3D structure from polyamide 2200 filled with hydrogels to facilitate the MSC reservoir and local immunosuppressant delivery of cytotoxic T lymphocyte-associated-protein 4 immunoglobulin (CTLA4Ig) [60]. The selection of the scaffold structure and composition depends on their purpose. For instance, chitosan hydrogel and scaffold structure delivery systems support exosomes derived from MSCs by improving their retention and stability [61]. A 3D Life Dextran-PEG Hydrogel is used to carry these cells and stimulate immunomodulatory effects on local tissues and infiltrate immune cells [62].

## 4. Applications and Mechanisms of Implantable Immunosuppressant Delivery to Prevent Allograft Rejection

Various types of immunosuppressants are used to prevent rejection after transplantation. Recent studies have reported innovative delivery systems to carry immunosuppressants successfully (Table 1 and Table 2).

### 4.1. Immunosuppressant Drugs

Immunosuppressants commonly utilized to treat patients who receive allotransplants, such as TAC [69], RAPA [70], and tofacitinib [23] have unique characteristics. Some of them are hydrophobic and others are quite hydrophilic. For example, TAC exhibits strong hydrophobicity, which impedes its applications in aqueous solutions. Although a previous study verified its successful encapsulation by micelles, its outcome remains unsuitable for further applications because of a low encapsulation efficiency [71].

For this reason, potential candidates, including thermosensitive polymer hydrogels, have been developed for TAC delivery. Thermosensitive polymer hydrogels exhibit excellent biocompatibility and biodegradability, have a low concentration requirement for gelation, and rapidly react to temperature change [72]. Chu et al. synthesized a thermosensitive polypeptide hydrogel with P-Lys-Ala-PLX (poloxamer and poly(l-alanine) with l-lysine segments at both ends) copolymers [73] to encapsulate TAC [39]. Poly(l-alanine) acts as the hydrophobic chain of P-Lys-Ala-PLX copolymers, which have been designed to capture the hydrophobic agents. A drug release kinetic study has shown that P-Lys-Ala-PLX hydrogel releases TAC drugs at a constant rate, indicating that this hydrogel type is appropriate for negatively charged proteins and drugs. A mixed hydrogel is formulated by adding poloxamers (Pluronic^®^ F127) to increase drug encapsulation efficiency and drug release rates. Furthermore, a supplement of methoxy-poly(ethylene glycol)-co-poly(lactic acid)-poly(ε-caprolactone) (mPEG-PLCL) and 0.5% polyvinylpyrrolidone (PVP) improve the stability of the hydrogel such that the drug was released in a stable and sustained fashion for 30 days. The allograft outcome was improved without significantly affecting systemic immune responses [65]. Another polymer that can also be loaded with TAC is the hydrogelator-triglycerol monostearate (TGMS), which also serves as a drug depot to slowly release drugs at a constant rate for long periods and exhibits enzyme-responsive properties. With these innovative delivery systems, the immunosuppressive effects of TAC were demonstrated by suppressing the local immune response, such as low complement C3c deposition, and proinflammatory cytokines (interleukin (IL)-2, tumor necrosis factor (TNF)-α, interferon (IFN)-γ, and IL-1-β). The development of chimerism in those hindlimb transplant recipients was also revealed [37,38,63].

Tofacitinib, the immunosuppressant used in the heart allograft, is fairly hydrophilic, indicated by a lower measured octanol-water partition coefficient (logP < 1.15) and is likely to cause rapid burst release. Thus, materials that contain the microcrystalline deposits of drugs encapsulated within an injectable hydrogel matrix can slow down the release rate. Therefore, hydrogel matrices allow drugs to stay localized, but early allograft rejection occurs when the location of administration is too far from the allograft [40].

The recent successful nanoparticle, micelle, and microsphere application for local immunosuppressant delivery systems in transplantation are listed (Table 2). For instance, a RAPA-nanomicelle that is used as an ophthalmic solution promoted the recruitment of myeloid-derived suppressor cells and improved their immunosuppressive function through the upregulation of arginase-1 and inducible nitric oxidase [56].

### 4.2. Biological Immunosuppressant

Antibodies and peptides that block inflammatory cytokines and interleukins are good candidates to prevent rejection. Naturally derived hydrogels, such as photocross-linkable gelatin-methacryloyl (GelMA)-based hydrogels, have been used for the controlled release of anti-IL-6 for skin allografts. Anti-IL-6 is locally released, as indicated by its local effects, such as less severe inflammation, nonoccurrence of matrix accumulation in draining lymph nodes, and unchanged systemic immune responses [64]. Another example, adding biological immunosuppressants, such as multimeric Fas ligand, in the hydrogel delivery system, improves MSCs survival by decreasing the CD8^+^ cytotoxic T cell population [26]. On the other hand, the microsphere system has been successfully developed to deliver an anti-TNF-α and IL-1-β for murine heart allografts [74]. Microspheres are phagocytosed by macrophages or other antigen-presenting cells [75]. This process verifies that the produced inflammatory cytokines become neutralized once they are released from macrophages. Unlike hydrogels or scaffolds, nanoparticles, microspheres, micelles, and liposomes may be administered through circulatory systems. To enhance site-specific targeting, conjugation of cell-specific ligands (including antibodies or specific antigens) to the surface of such molecules is a promising strategy [76]. The conjugation of endothelial-specific molecules improves site-specific targeting of endothelium, for example, platelet–endothelial cell adhesion molecule-1 increases lung targeting in vivo [77,78]. In transplantation, the conjugation of donor antigens specific to nanoparticles has been developed to prolong allograft survival without affecting overall recipient immune function [45,67,79,80].

## 5. Efficacy and Efficiency Compared to Conventional Administration

Efficacy and efficiency are very crucial for drug administration. Although the conventional immunosuppressant administration has resulted in the inhibition of the alloimmune response, the dosage may not be optimal. Without the help of biomaterials that can sustain long-term drug release, their effects in recipients will not be long lasting [81]. Therefore, a daily injection is required that results in a large amount of immunosuppressant dose. Although the physician maintains the systemic dose, it is still not proportionally correlated with local allograft concentrations. Furthermore, significant interpatient and intrapatient variability in drug exposure due to various clinical conditions, such as systemic inflammation, hemorrhage, and shock, often occurs [82]. In consequence, several problems exist in the conventional application of immunosuppressants. First, most immunosuppressants that are administered systematically may induce severe side effects. Fluctuating drug absorption will disrupt protein metabolism and cause loss of (multi) organ function. In the early phase after transplantation, it results in subsequent nephrotoxicity [82]. In addition, it increases the risk of infection [83] or cancer [84]. Second, a high concentration of immunosuppressants causes toxicity in stem cells [85]. Third, noncompliance by the patient during long-term medication fails to ensure the good efficacy of immunosuppressants [86].

To overcome these problems, localized and sustained immunosuppressant delivery is urgent. Implantable immunosuppressant delivery may be a safer route for drug administration. The several advantages of local immunosuppressant delivery compared to systemic delivery are shown in Table 3. With the incorporation of immunosuppressants into the local delivery system, such as hydrogels, the local alloimmune response has been greatly suppressed without systemic side effects. Several novel immunosuppressant-delivery systems that could improve the effectiveness of immunosuppressants have been reported. With a single injection of the immunosuppressant-loaded local delivery system, it can sustain the release of the immunosuppressant for one month or more, depending on the initial dose that is administered on day zero. Although burst release occurs in the early period after injection, the immunosuppressant was slowly and stably released for the long term. Moreover, that single injection has been shown, not only to improve allograft outcome, but also preserve kidney function without occurrence of malignancies or opportunistic infections compared with systemic immunosuppression [37,38,63,64,87,88]. With these advantages, the problem of noncompliance in the patient can be overcome [89,90]. However, additional costs are required for these implantable immunosuppressant deliveries [91,92].

## 6. Biocompatibility and Safety

High biocompatibility of the novel immunosuppressant-delivery systems is indicated by their high flexibility to interact with the cell surface or any protein in the body. Biocompatibility can be tested in vitro or in vivo. In vitro tests cannot replace in vivo tests but can minimize the animal sacrifice [93]. To standardize the protocol for biocompatibility, the manufacturer should follow the International Organization for Standardization (ISO) 10993 guidelines [94]. The recent use of polymeric biomaterials in transplantation studies, such as PLGA, PCL, and PEG, showed good biocompatibility, including cellular response, cell adhesion, and hemocompatibility [95,96]. Biocompatibility may be further enhanced by modification or addition of other polymeric materials [97,98,99,100]. For those nanoparticles, the ISO standard 1099-22 test that is dedicated to assessing the impact of shape, size, surface charge and function, dispersion state, matrix composition, attached protein formation on cell viability, and endocytosis process have been described [101,102]. The delivery drugs are not only polymeric with high biocompatibility but can also be applied to the niche of the stem cells, as shown by the thermosensitive chitosan hydrogel scaffold with microstructures or nanostructures [103,104].

Testing the safety of biomaterials usually includes a sensitization test, an irritation test in animals/humans, and a cytotoxicity assay both in the in vitro and in vivo systems [105]. In vitro studies have been performed with the relevant cultured cell line, such as human embryonic kidney cells [39], endothelial progenitor cells [41], or specific cell lines for certain experiments (e.g., human corneal epithelial cells for cornea transplantation) [106]. In the in vivo study, toxicity has been investigated by measuring parameters of kidney and liver function, such as blood urine nitrogen, creatinine [41], cholesterol, triglycerides, and liver enzymes, such as aspartate aminotransferase and alanine aminotransferase [37,107]. Polymeric materials that have been utilized in immunosuppressant micelle and nanoformulations have been extensively studied for decades. The administration of PLGA CsA [108], PEG-chitosan CsA nanoparticles (NPs) [109], poly(ethylene oxide-b-ε-caprolactone) (PEO-b-PCL) CsA micelle [110], and albumin-based TAC NPs [111] intravenously showed no kidney damage or safety enhancement. In the recipients who received TAC-loaded TGMS hydrogels, all these values have been lower than in those who received systemic TAC [37].

## 7. Conclusions

The unique properties of implantable immunosuppressant delivery, such as hydrogels, micelles, and microspheres, are suitable to avoid rapid immunosuppressant release. The ability to transform from solution into a solid form makes hydrogels excellent platforms for localized drug delivery applications. The nanosize of the nanoparticle will enhance the absorption by the targeted organ and the antigen-specific modification will improve their site-specific targeting. In addition, the toxicity studies on these delivery systems have validated their safety. The current literature review of the applications of various implantable delivery systems, including hydrogel-based (thermosensitive, photosensitive, and enzyme responsive), modified hydrogel-based systems with scaffolds and peptides, and others (micelles, microsphere) for the delivery of immunosuppressants to allografts, demonstrates that implantable immunosuppressant delivery systems provide many advantages, such as limited toxicity in normal tissues, the localized and sustained delivery of the immunosuppressants in the allograft vicinity, improved stability of the immunosuppressants, increased efficiency of the immunosuppressant effect, and enhanced allograft acceptance compared with systemic immunosuppressant administrations.

## 8. Challenge and Future Perspective

The effectiveness of these delivery systems for localized immunotherapy has been well characterized with standard in vitro environment protocols. The proof-of-principle demonstrations of their activities in various rodent–porcine allograft models are also encouraging. Although many of the in vivo studies exploring hydrogels have been extensively conducted, studies on nonhuman primates are still required. Moreover, necessary validation of the biocompatible and biodegradable polymers as immunosuppressant-delivery treatment options should be addressed in clinical studies to assess the effectiveness of these drug delivery platforms for allograft treatment.

Although the burst release pattern of immunosuppressants after injection is still below the maximum safety dose, modifications may be required to avoid incidence. Furthermore, the site-specific immunosuppression offered by this innovative delivery system is an effective strategy to achieve favorable results without the side effects of systemic immunosuppression. This provides an attractive option, not only in organ transplantation, but also in vascularized composite allotransplantation, where the transplanted allograft is easily visible.

## Figures and Tables

**Figure 1 ijms-23-01592-f001:**
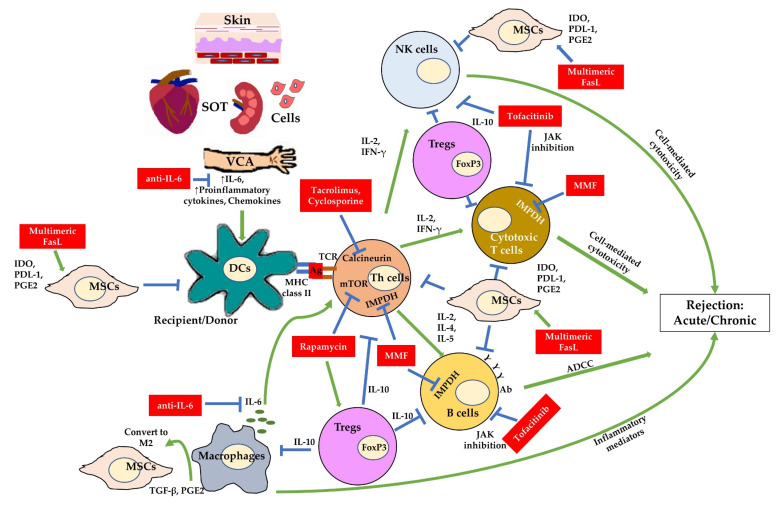
Examples of immunosuppressant targets in allograft rejection. ADCC: antibody-dependent, cell-mediated cytotoxicity; Ab: antibody; Ag: antigen; DC: dendritic cells; FasL: Fas ligand; FoxP3: forkhead box P3; IMPDH: inosine monophosphate dehydrogenase; IDO: indoleamine 2,3-dioxygenase 1; IFN-γ: interferon gamma; IL: interleukin; JAK: janus kinase; MHC: major histocompatibility complex; MMF: mycophenolate mofetil; MSCs: mesenchymal stem cells; mTOR: mammalian target of rapamycin; natural killer cells: NK cells; PDL-1: programmed cell death-1; PGE2: prostaglandin E2; SOT: solid organ transplantation; TGF-β: transforming growth factor beta; Tregs: regulatory T cells; VCA: vascularized composite allotransplantation.

**Figure 2 ijms-23-01592-f002:**
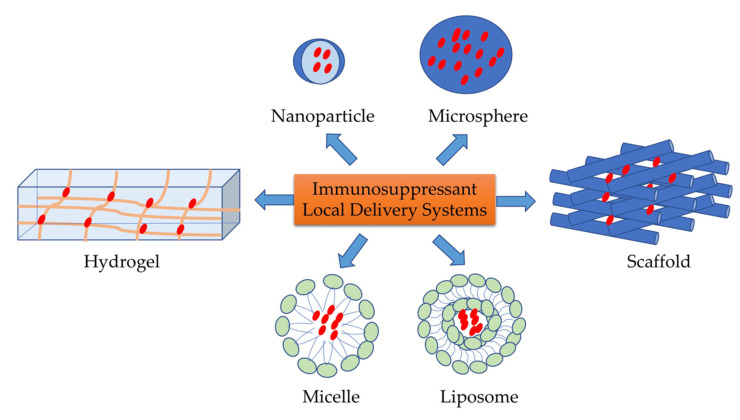
Types of local delivery systems.

**Table 1 ijms-23-01592-t001:** Recent hydrogel application as immunosuppressant delivery to prevent allograft rejection.

Literature, Year	Delivery Systems	Polymer	Transplantation	Animal Model	Immunosuppressant, Dose	Outcome (Survival Rate)
Gajanayake et al., 2014 [38]	Enzyme responsive hydrogel	TGMS	VCA (Orthotopic hindlimb)	Rat	TAC, 7 mg	6/6, >100 days
Dzhonova et al., 2018 [37]	Enzyme responsive hydrogel	TGMS	VCA (Orthotopic hindlimb)	Rat	TAC, 7 mg per 70 days	5/6, >280 days
Li et al., 2018 [41]	Nanoparticle-anchoring hydrogel scaffold	PLGA-PEG-maleimide	Stem cells (EPCs)	Mouse	TAC, 1 mg	12/12, >14 days
Fries et al., 2019 [63]	Enzyme responsive hydrogel	TGMS	VCA (Orthotopic forelimb)	Porcine	TAC, 49 mg	4/4, 56–93 days
Uehara et al., 2019 [64]	Photocrosslinkable hydrogel	Gelatin- methacryloyl	Skin	Porcine	Anti-IL-6, (0.4% *w*/*v*)	5/5, MST: 23 days
Majumder et al., 2020 [40]	Microcrystalline hydrogel ^1^	TFA/ thioanisole/ ethanedithiol	Heart	Mouse	Tofacitinib, 750 μg	5/5, up to 160 days
Wu et al., 2021 [65]	Thermosensitive hydrogel	mPEG-PLCL/PVP	Skin	Rat	TAC, 10 mg	2/6, >30 days
Alvarado-Velez et al., 2021 [26]	Lipid microtubes agarose hydrogel	Agarose	MSC	Rat	Multimeric Fas ligand 12 μg	8/8, >6 days

^1^ Additional treatment with CTLA4Ig was applied. EPCs: endothelial progenitor cells; mPEG-PLCL: methoxy-poly(ethylene glycol)-co-poly(lactic acid)-poly(ε-caprolactone); MSC: mesenchymal stem cells; MST: mean survival time; PEG: polyethylene glycol; PLGA: poly-lactic-co-glycolic acid; PVP: polyvinylpyrrolidone; TFA: trifluoroacetic acid; TGMS: triglycerol monostearate; TAC: tacrolimus; VCA: vascularized composite allotransplantation.

**Table 2 ijms-23-01592-t002:** Recent microsphere and nanoparticle application as immunosuppressant delivery systems to prevent allograft rejection.

Literature, Year	Delivery Systems	Polymer	Transplantation	Animal Model	Immunosuppressant, Dose	Outcome (Survival Rate)
Wang et al., 2015 [45]	Microsphere	PLGA	Skin	Mouse	H-2K^b^/OVA257–264 monomers, 100 μg	16/16, 14–19 days
Unadkat et al., 2017 [46]	Disk microsphere	PLGA	VCA (Orthotopic hindlimb)	Rat	TAC, 40 mg	6/6, >180 days
Wei et al., 2018 [56]	Nanomicelles	PVCL-PVA-PEG	Cornea	Mouse	RAPA, 3 × 0.05 mg/day	18/20, >60 days
Kuppan et al., 2019 [66]	Micelles ^1^	PLGA	Islet	Mouse	Dexametasone, 2 mg	8/10, >60 days
Liu et al., 2019 [57]	Nanomicelles	NH2-PEG-b-PLA and mPEG-b-PLA	Cornea	Rat	TAC, 87.5 μg/day	10/10, MST: 27.5 days
Shah et al., 2019 [67]	Nanoparticles	PLG	Skin	Mouse	ECDI peptide, 3 mg	6/6, >90 days
Wang et al., 2020 [44]	Microsphere	PLGA	VCA (Orthotopic hind-limb)	Rat	TAC, MMF, PDNN, 6 mg, 300 mg, 60 mg	6/6, >150 days
Deng et al., 2021 [68]	Nanoparticles	PLGA	Heterotopic abdominal heart	Rat	TAC, 3 mg/kg	3/6, >28 days

^1^ Additional treatment with CTLA4Ig was applied. ECDI: 1-ethyl-3-(3’-dimethylaminopropyl)-carbodiimide; mPEG: methoxy-poly(ethylene glycol); MMF: mycophenolate mofetil; PDNN: prednisolone; PEG: polyethylene glycol; PLA: poly(d,l)-lactic acid; PLG: polylactide-co-glycolide; PLGA: poly-lactic-co-glycolic acid; PVA: polyvinyl alcohol; PVCL: polyvinyl chloride; RAPA: rapamycin; TAC: tacrolimus; VCA: vascularized composite allotransplantation.

**Table 3 ijms-23-01592-t003:** Comparison between systemic and local immunosuppressant administration.

	Systemic Administration	Local Delivery	Ref
Concentration in the blood	Fluctuate	Initial burst then stable	[38,63,82,87]
Side effects	More	Less	[37,82,84,87,88,89]
Patient compliance issue	More	Less	[86,89,90]
Dosing frequency	High	Low	[37,38,64,81,89]
Drug efficacy	Low	High	[38,63,81]
Drug dosage	More	Less	[38,63,81]
Additional cost	No	Yes (with biomaterial production)	[91,92]

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
