# Peer review of "Implantable Immunosuppressant Delivery to Prevent Rejection in Transplantation"

_ijms, 2022, doi:10.3390/ijms23031592_

Round 1

Reviewer 1 Report

Well presented. No major corrections/revisions required

Author Response

  1. Well presented. No major corrections/revisions required

Author Response:

Thank you for your supportive comments. The revised manuscript has been proofread by native English speaker with expertise in the field. All changes in the manuscript are highlighted in red.

Reviewer 2 Report

Recipients of allograft transplantation often require long-term immunosuppressive treatment to prevent graft rejection. To extend the effective entity of the immunosuppressants and to achieve a better controlled drug delivery, multiple innovative drug delivery systems have emerged in the biomedical field in recent years. The authors aim to summarize studies that used implantable immunosuppressant delivery to release therapeutic, and to compare their applications, efficacy, efficiency, and safety with conventional drug administration.

My comments are as follow:

  1. The sections about innate and adaptive immune responses are too vague for readers in biomedical but also too unclear for the general public. Since the statements in these sections are not accurate enough and didn’t form a strong link to the rest of the manuscript, it’s better to minimize this part and provide a couple good and up-to-date references for readers that need basic immunology background to go through.
  2. Avoid using the word “graft” when referring to the host cell/tissue/organ because it is confusing in an article about transplantation.
  3. Figure 1 is not accurate and requires improvement. For example, the cellular responses can cause rejection directly without going through humoral responses. The immunosuppressants also can act on other immune components other than T cells. It’s also rarely used in transplant immunology the term “acceptance”, but usually “tolerance”. The figure also completely ignores the innate immunity although the legend is generalized as “Transplantation and Immune Response”. If the authors could not generate a graph that contains correct and complete message on immune mechanisms of allograft rejection, it is better to not showing it, but refer the readers to other detailed reviews or book chapters that specifically discuss these immune mechanisms, and focus this review on its topic, which is the novel drug delivery used in transplant settings.
  4. Please make sure to provide full term for any abbreviation appeared the first time in the article.
  5. Since the review is about these drug delivery system in transplant field, the authors might want to provide more insight into the outcome and conclusion from those cited studies. So instead of listing them out, more detail from these studies should be discussed.
  6. The implantation of hydrogel or scaffold for local drug release is obvious, but not the nanoparticles or liposomes as they would be sent into circulation. Therefor the studies that involved design of these delivery systems to target specific tissue or organ should be mentioned and discussed.

Author Response

Thank you very much for reviewing our manuscript entitled “Implantable Immunosuppressant Delivery to Prevent Rejection in Transplantation” submitted for consideration for publication in International Journal of Molecular Sciences.

Reviewers have mentioned stimulating and constructive comments that were very beneficial to revise and improve this review. As suggested, we have addressed all their concerns, and all changes in the manuscript are highlighted in red.

The point-to-point response to the Reviewers’ comments as follows:

  1. The sections about innate and adaptive immune responses are too vague for readers in biomedical but also too unclear for the general public. Since the statements in these sections are not accurate enough and didn’t form a strong link to the rest of the manuscript, it’s better to minimize this part and provide a couple good and up-to-date references for readers that need basic immunology background to go through.

Author Response:

Thank you for your opinion. To improve the readability, the section about innate and adaptive immune responses has been revised. The revised manuscript has been proofread by native English speaker with expertise in the field. (Page 2 line 59)

  1. Avoid using the word “graft” when referring to the host cell/tissue/organ because it is confusing in an article about transplantation.

Author Response:

Thank you for your suggestion. Word “graft” has been replaced. (Page 1 line 44, Page 2 line 46)

  1. Figure 1 is not accurate and requires improvement. For example, the cellular responses can cause rejection directly without going through humoral responses. The immunosuppressants also can act on other immune components other than T cells. It’s also rarely used in transplant immunology the term “acceptance”, but usually “tolerance”. The figure also completely ignores the innate immunity although the legend is generalized as “Transplantation and Immune Response”. If the authors could not generate a graph that contains correct and complete message on immune mechanisms of allograft rejection, it is better to not showing it, but refer the readers to other detailed reviews or book chapters that specifically discuss these immune mechanisms, and focus this review on its topic, which is the novel drug delivery used in transplant settings.

Author Response:

Thank you for your concern. We would like to keep the Figure 1 in the manuscript to let the reader understand the importance of immunosuppressants in transplantation. The figure has been revised and the caption of figure has been edited with the more appropriate one. (Page 3 line 83)

  1. Please make sure to provide full term for any abbreviation appeared the first time in the article.

Author Response:

Thank you for your suggestion. Full terms of all abbreviations have been provided. (Page 2 line 56, 59, 73, 76, Page 5 line 155, Page 7 line 211-212)

  1. Since the review is about these drug delivery system in transplant field, the authors might want to provide more insight into the outcome and conclusion from those cited studies. So instead of listing them out, more detail from these studies should be discussed.

Author Response:

Thank you for your valuable suggestion. The mechanisms, outcome, and conclusion of the cited studies have been discussed. (Page 7 line 180 to Page 8 line 255)

  1. The implantation of hydrogel or scaffold for local drug release is obvious, but not the nanoparticles or liposomes as they would be sent into circulation. Therefor the studies that involved design of these delivery systems to target specific tissue or organ should be mentioned and discussed.

Author Response:

Thank you for your precious advice. Studies that involved the design of these delivery systems to target a specific tissue or organ have been mentioned and discussed. (Page 9 line 291)

Round 2

Reviewer 2 Report

Although some improvement has been made, several issues still exist. Some of the issues are:

  1.  
  2. The section 2 titled “Immunosuppressant to Prevent Allograft Rejection in Transplantation”, however, the entire text under this section did not focus on this. Instead, it selectively describes fundamental immunology without going into any immunosuppressant, except for showing some of them in the figure 1.
  3. Repetition between lines 111-113 (“Addition of a self-assembled peptide (RADA16) as a basic unit of hydrogels could anchor and immobilize the nanoparticles therefore can localize and sustain drug release for 28 days [34].”) and lines 280-283 (“Addition of peptide, such as RADA16, to the hydrogel has prolonged the immunosuppressive effects for a long time for more 21 days, shown as pro-inflammatory cytokines (IFN-γ, IL-2, and IL-6) reduction [34].”) while the effective length were different.
  4. The paragraph between lines 276-290 doesn’t have a clear purpose and a logical connection between sentences.
  5. Specific references/citations should be given on Table 3.
  6. Lines 241-245, please clarify “is fairly hydrophilic with a measured logP <1.15”.
  7. Line 248, “exosome-derived MSCs” is clearly a mistake.

Author Response

Dear Reviewer,

Thank you very much for your comprehensive comments on our manuscript entitled “Implantable Immunosuppressant Delivery to Prevent Rejection in Transplantation”.

We feel we have satisfactorily responded to your concerns, and are resubmitting our manuscript with a point-by-point response to the comments. Note that all changes in the text have been noted in red color.

The point-to-point response to your comments as follows:

1. The section 2 titled “Immunosuppressant to Prevent Allograft Rejection in Transplantation”, however, the entire text under this section did not focus on this. Instead, it selectively describes fundamental immunology without going into any immunosuppressant, except for showing some of them in the figure 1.

Author Response:

Thank you for your opinion. We aim to make the readers understand the importance of immunosuppressants in transplantation by illustrating the basic concepts of transplantation immunology. The section has been revised again to support the title as follows:

“Transplantation is featured to transfer an organ, a tissue, or a group of cells from the donor to the recipient (or host) or to different sites in the same individual [1], with the purpose to save lives or improve the quality of life by replacing devastating organ or tissue with the functional one [2,3]. However, rejection caused by the direct or indirect immune mechanism of the recipient, is one of the main obstacles that occur after transplantation [4].

In a transplantation setting, both innate [5] and adaptive immune cells [6,7], such as macrophages, natural killer (NK) cells, helper T cells (Th) (CD4+ T cells), cytotoxic T cells (CD8+ T cells), B cells, as well as cytokines and antibodies are involved in direct or indirect immune mechanism of allograft rejection [8]. To inhibit effector cell activation/proliferation or enhance regulatory cell activation/proliferation [9], the common approaches to immunosuppression include costimulation blockades to block dendritic cells (DC)-effector T cell interactions [10], anti-inflammatory recombinant proteins [11,12], immunosuppressive drugs [13], and anti-inflammatory cytokines derived from immunomodulatory stem cells, such as mesenchymal stem cells (MSCs) [14,15].

The early development of immunosuppressants, such as cyclosporine (CsA), tacrolimus (TAC), and rapamycin (RAPA), were targeting T cells pathway [16]. This is because recipient T cells play a significant role in acute/chronic rejection [17] with direct recognition of foreign antigens that are presented to donor and recipient dendritic cells by their receptor [18,19]. Once T cells are activated, they proliferate and secrete cytokines that regulate other immune responses, including B cells [20] that can produce an alloantibody that is detrimental to long-term graft survival [21]. Mycophenolate mofetil (MMF) has been applied to inhibit B cell proliferation [22]. Later on, more immunosuppressant types were developed. For instance, tofacitinib that inhibits the immune response through the janus kinase pathway, monoclonal antibodies (clazakizumab, tocilizumab) to inhibit inflammatory cytokines [such as interleukin (IL)-6], and multimeric Fas ligand (Adipogen) that can improve the survival of MSCs survival and support MSCs to induce T cell apoptosis (Figure 1) [23-27].

Broadly speaking, the transplant recipients may receive pharmacological treatments during the induction, maintenance, or rejection phases [28,29] based on the posttransplant time and the specific infection/rejection profile status [13,30]. Treatment during the induction phase aims to immediately eradicate the immune response, while treatment during the maintenance phase seeks to maintain the tolerant condition in which the immune system is sufficiently suppressed to minimize allograft rejection, but still able to respond to infections [31]. Multi- or single-immunosuppressive drug administration is commonly applied in the induction or maintenance phases. Although not always successful, combination therapy that is intended to deplete B cells is used in the rejection phase. Costimulation blockades, T cells or lymphocytes depletion, and immunomodulatory cells are also used during the induction phase because immunosuppressive drug administration alone is not enough [32].”

2. Repetition between lines 111-113 (“Addition of a self-assembled peptide (RADA16) as a basic unit of hydrogels could anchor and immobilize the nanoparticles therefore can localize and sustain drug release for 28 days [34].”) and lines 280-283 (“Addition of peptide, such as RADA16, to the hydrogel has prolonged the immunosuppressive effects for a long time for more 21 days, shown as pro-inflammatory cytokines (IFN-γ, IL-2, and IL-6) reduction [34].”) while the effective length were different.

Author Response:

Thank you very much for your review in details. According to our understanding of the cited articles, we wrote the lines 111-113 (now lines 148-151) to describe how long the drug release/exist but lines 280-283 describes that those observation for pro-inflammatory cytokines (IFN-γ, IL-2, and IL-6) reduction was held up to 21 days. We deleted the sentence in line 290-293 and revised the section 5 to avoid the repetitive sentences and ambiguity.

3. The paragraph between lines 276-290 doesn’t have a clear purpose and a logical connection between sentences.

Author Response:

Thank you for your comments. The purpose is to compare the effectiveness of drug delivery administration between the conventional method and the novel approach. We tried to rewrite the paragraph with the help of a professional native English speaker. To improve the clarity and readability, Section 5 has been revised as follows:

Line 246-280

“Efficacy and efficiency are very crucial for drug administration. Although the conventional immunosuppressant administration has resulted in the inhibition of the alloimmune response, the dosage may not be optimal. Without the help of biomaterials that can sustain long-term drug release, their effects in recipients will not be long-lasting [81]. Therefore, a daily injection is required that results in a larger amount of immunosuppressant dose. Although the physician maintains the systemic dose, it is still not proportionally correlated with local allograft concentrations. Furthermore, significant interpatient and intrapatient variability in drug exposure due to various clinical conditions, such as systemic inflammation, hemorrhage, and shock, often occurs [82]. In consequence, several problems exist in the application of immunosuppressants conventionally. First, most immunosuppressants that are administered systematically may induce severe side effects. Fluctuating drug absorption will disrupt protein metabolism and cause loss of (multi-) organ function. In the early phase after transplantation, it results in subsequent nephrotoxicity [82]. In addition, it increases the risk of infection [83] or cancer [84]. Second, a high concentration of immunosuppressants gives toxicity to stem cells [85]. Third, non-compliance by the patient during long-term medication fails to ensure the good efficacy of immunosuppressants [86]. To overcome these problems, localized and sustained immunosuppressant delivery is in urgent. Implantable immunosuppressant delivery may be a safer route for drug administration. The several advantages of local immunosuppressant delivery compared to systemic delivery are shown in Table 3. With the incorporation of immunosuppressants into the local delivery system, such as hydrogel, the local alloimmune response has been greatly suppressed without systemic side effects. Several novel immunosuppressant delivery systems that could improve the effectiveness of immunosuppressants have been reported. With a single injection of the immunosuppressant loaded local delivery system, it can sustain the release of immunosuppressant for one month or more, depending on the initial dose that is administered on day 0. Although burst release occurs in the early period after injection, the immunosuppressant was slowly and stably released for the long term. Moreover, that single injection has been shown not only improve allograft outcome but also with preserved kidney function, without occurrence of malignancies or opportunistic infections, compared with systemic immunosuppression [37,38,63,64,90]. With these advantages, the problem of noncompliance in the patient can be overcome [87,89]. However, additional cost is required for these implantable immunosuppressant delivery [91,92].”

4. Specific references/citations should be given on Table 3.

Author Response:

Thank you for your suggestion. Specific references/citations have been given on Table 3.

5. Lines 241-245, please clarify “is fairly hydrophilic with a measured logP <1.15”.

Author Response:

Thank you for your advice. According to our understanding, logP, known as the octanol-water partition coefficient, is a measurement value that indicates how hydrophilic or hydrophobic a molecule is. The lower logP (the value can even be negative) means that the molecule has hydrophilic behavior, while the higher logP means that the molecule has hydrophobic characteristic. <1 logP is considered as hydrophilic, 0-3 logP is considered as moderate hydrophilic, and >5 logP is considered as highly hydrophobic(1, 2).

To improve clarity, the sentence has been revised as shown in lines 213-214.

“ is fairly hydrophilic, indicated by a lower measured octanol-water partition coefficient (logP <1.15)”

6. Line 248, “exosome-derived MSCs” is clearly a mistake.

Author Response:

Thank you for your correction. The word has been corrected to “exosome derived from MSC”.

(line 156)

References

  1. Takacs-Novak K. Physico-Chemical Methods in Drug Discovery and Development. Mandic E, editor. Zagreb, Croatia: International Association of Physical Chemists (IAPC) Publishing; 2012.
  2. Dashti Y, Grkovic T, Quinn RJ. Predicting natural product value, an exploration of anti-TB drug space. Nat Prod Rep. 2014;31(8):990-8.

Sincerely,

Authors

Round 3

Reviewer 2 Report

The authors have made significant improvement to the manuscript.

Minor suggestions:

  1. Line 58, .... that are presented "by" donor and recipient dendritic cells. (Instead of "to").
  2. Figure 1, consistency in singular or plural forms in the graph and legend.
  3. Table 3, align text position for each row. Additionally, the plus and minus symbols placed at "side effects" and "patient compliance" are confusing --systemic delivery has problem with side effects but not with patient compliance, while local delivery would have no problem with side effects but have issue with patient compliance?    

Author Response

Dear Reviewer,

Thank you very much for re-examining our manuscript entitled “Implantable Immunosuppressant Delivery to Prevent Rejection in Transplantation”.

We feel we have satisfactorily responded to your concerns, and are resubmitting our manuscript with a point-by-point response to the comments. Note that all changes in the text have been noted in red color.

The point-to-point response to your comments as follows:

The authors have made significant improvement to the manuscript.

Minor suggestions:

  1. Line 58, .... that are presented "by" donor and recipient dendritic cells. (Instead of "to").

Author Response:

Thank you for your correction. We have replaced “to” with “by”.

  1. Figure 1, consistency in singular or plural forms in the graph and legend.

Author Response:

Thank you for your reminder. We have revised the inconsistent words in the figure, graph and legend, for instance:

MSC has been changed to MSCs; Treg cell --> Tregs; B cell --> B cells. Th cell and Cytotoxic T cell have also been replaced with Th cells and Cytotoxic T cells, respectively.

  1. Table 3, align text position for each row. Additionally, the plus and minus symbols placed at "side effects" and "patient compliance" are confusing --systemic delivery has problem with side effects but not with patient compliance, while local delivery would have no problem with side effects but have issue with patient compliance?    

Author Response:

We really appreciate your detailed review. We have aligned text position for each row. Furthermore, we agree that those marks are confusing. We would clarify that local delivery would have less problem with side effects but patient compliancecould be accomplished. We have now removed the marks and replaced with appropriate words to clarify the meaning.

Sincerely,

Authors
